# Hip stabilization in an australopithecine-like hip: the influence of shape on muscle activation

Patricia Ann Kramer[1],* and Adam D. Sylvester[2]

**ABSTRACT**

Hip stabilization through muscular activation of the gluteals is a key feature of hominin walking, but the role of pelvic shape on muscular activation remains uncertain. Coupled with this is the uncertainty regarding whether the kinematics and kinetics of modern humans are appropriate in extinct hominins. We apply modern human kinematics and kinetics to musculoskeletal models with modern human-like and australopithecine-like hips. We test the prediction that the hip functional complex that includes biacetabular breadth, femoral neck length, and iliac blade flare, produces hip abductor muscle activations that are similar in the modern human- and australopithecine-like forms. Using previously developed musculoskeletal models, we calculated muscle forces using inverse dynamics analyses and a muscle redundancy algorithm for ten individuals who walked at their normal velocity. We found that the shape of the australopithecine-like pelvis produces absolutely higher muscle activations in gluteus medius and gluteus minimus, but lower muscle activations across a long period of stance in gluteus maximus compared to the modern human-like pelvis when kinematics and size are held constant. These results suggest that, while the australopithecine-like pelvis is compatible with human walking patterns, influences on pelvic shape other than accommodating muscle and joint reaction forces during walking are present.

KEY WORDS: A.L. 288-1, Musculoskeletal modeling, Pelvis shape

**INTRODUCTION**

As one of the foundational transitions in human evolution, understanding the adaptive origin of bipedalism remains a critical task in anthropology and is one that ultimately depends on elucidating the details of earlier forms in our lineage. Based on fossilized skeletal material, various researchers have argued that most, if not all, extinct hominins practiced some form of bipedalism that was kinematically, kinetically, and/or metabolically distinct from modern bipedalism. These groups include *Ardipithecus ramidus* (Lovejoy et al., 2009), *Australopithecus africanus* (Ruff and Higgins, 2013), *Paranthropus robustus* (Ruff and Higgins, 2013), *Australopithecus afarensis* (Bates et al., 2025; Kramer, 1999;

O'Neill et al., 2024; Ruff and Higgins, 2013; Wiseman, 2023), *Australopithecus sediba* (DeSilva et al., 2013), *Homo habilis* (Wood and Collard, 1999), *Homo naledi* (Walker et al., 2019), *Homo erectus* (Simpson et al., 2008), and Neandertals (Been et al., 2011; Vidal-Cordasco et al., 2017). Much of this research has focused on the skeletal anatomy of the lower limb due to the critical role these structures play in weight-bearing and locomotion. The hip (i.e. pelvis and proximal femur) has been of particular interest because features of these skeletal elements, such as anteriorly wrapped iliac blades (e.g. Lovejoy, 1988) and asymmetrical cortical thickness in the femoral neck (e.g. Lovejoy et al., 2002; Ruff and Higgins, 2013), are universally agreed to indicate bipedalism.

The differences in morphology between australopithecines (we refer to this morphology as 'australopithecine-like'), of which A.L. 288-1 provides a mostly complete pelvis and modern humans ('modern human-like') can be categorized into issues of size and shape. A.L. 288-1 is generally considered to be small [e.g. with a body mass of less than 35 kg (Ruff and Wood, 2023; Ruff et al., 2018), with a stature of 1.1 m (Doran, 1997; Hens et al., 2000; Jungers, 1982) and a femoral length of 0.277 m (Sylvester et al., 2008)], but with a biacetabular breadth that is relatively large compared to stature (Tague and Lovejoy, 1986). This pairing of small mass, stature, and femur length with a wide pelvis is outside the range of variation of modern humans. In addition to differences in size, the shapes of the pelvis and femur of A.L. 288-1 are different from that of modern humans, but similar to other australopithecines like *A. africanus* (e.g. Häusler and Schmid, 1995; Ruff and Higgins, 2013). For example, the iliac blades of A.L. 288-1 are less anteriorly wrapped than in modern humans, resulting in a more posterior position of the anterior superior iliac spine (ASIS) and smaller area of attachment for the hip abductors anterior to the acetabulum (Johanson et al., 1982; Lovejoy, 1988; Stern and Susman, 1983) (Figs 1 and 2; Movie 1).

While early research connecting form and function was both typological and qualitative in nature, anthropologists have increasingly adopted mechanical approaches to answer questions about primate locomotion generally and hominin locomotion in particular (e.g. see a review by Ruff, 2018). These mechanical analyses allow the development of muscle and joint reaction forces, which are necessary to understand skeletal morphology. Bone shape responds to loading in the life of an individual (e.g. Frost, 1982, 2003; Wolff, 1892) while skeletal differences between species are mediated by natural selection (and/or other evolutionary forces). The form of a species (or other group) is, presumably, selected in the context of the species' functional requirements where 'typical' loadings (e.g. those that an individual would experience during species-typical movements) produces strains that are in the 'adapted window' of bone (Frost, 2003). The skeletal morphology of any individual is, then, the product of its manifestation of the species form (Lovejoy et al., 1999) and the mechanical forces the individual experienced in its life. Particularly important when the form is

[1]Dept of Anthropology, University of Washington Seattle Campus: University of Washington, Box 353100, Denny Hall, Seattle, WA 98195-3100, USA. [2]Center for Functional Anatomy and Evolution, The Johns Hopkins University School of Medicine, 1830 E. Monument Street, Baltimore, MD 21205, USA.

*Author for correspondence (pakramer@uw.edu)

P.A.K., 0000-0002-6435-9130; A.D.S., 0000-0002-5234-074X

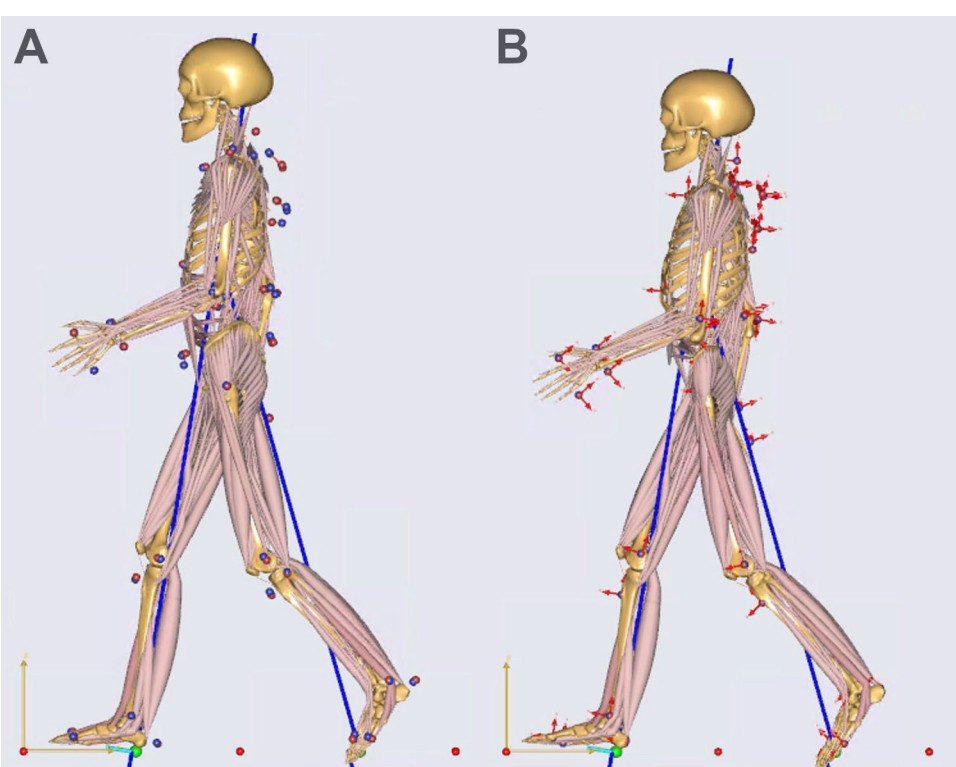

**Fig. 1. ADL modern human- (A) and ADL australopithecine-like (B) musculoskeletal models.** The same midstance frame is shown for both.

extinct and no direct observations can be made, virtually exploring the strain profiles of bone can help us understand why a morphology exists (e.g. Maggiano et al., 2008; Ruff, 2018) or, vice versa, what morphology a set of forces should produce (e.g. Shefelbine et al., 2002). Another reason for interest in muscular forces is that muscular activation uses energy: more active volume and/or higher forces use more energy (e.g. Beck et al., 2022; Ortega et al., 2015). Energy is a critical resource that once used for a task cannot be reused for another (Smith and Winterhalder, 1984). Locomotion, with its attendant energy use for muscle activation, is a critical task but so too are tasks like maintenance and reproduction. We start from the presumption that natural selection acts on musculoskeletal structures to produce forms that are sufficient for the individual to survive and reproduce and that the representation of characteristics that enable or enhance these basic functions proliferate over time (Kramer and Sylvester, 2009). Consequently, extinct forms should be examined from the perspective that they were adapted to their environmental circumstances given their evolutionary history. This means two things: species form is not 'compromised' or 'transitional' and individual variation within a species should exist within evolutionarily acceptable norms of reaction for any trait or function.

Within the anthropological literature (Heiple and Lovejoy 1991; Lovejoy et al., 1973) were the first to estimate hip joint reactions and femoral loads during walking based on a model adopted from orthopaedic biomechanics (Ruff, 2018). The model was adopted by Ruff (1995) and Lovejoy et al. (1973), among others, to estimate hip joint reaction forces and bending moments in femoral shafts. This simplified approach posed the question of hip biomechanics as a two-dimensional coronal plane, static problem. While this was perhaps a necessary simplification at the time, walking is inherently three-dimensional in terms of space with an additional temporal dimension. Critical information is potentially lost by projecting this four-dimensional problem into two dimensions.

Fortunately, understanding the neuromuscular basis for human movement is of great interest across a wide variety of research fields (e.g. sports performance, clinical gait) and advances in computing and algorithm development have led to sophisticated models of human anatomy [e.g. the MoCap model in the AnyBody Managed Model Repository (AMMR v2.3.0)]. Originally driven by a clinical interest in gait pathologies, musculoskeletal modeling has emerged as a technique to elucidate the details of human (and other animal) movement (Delp et al., 2007). These models build upon forward and inverse dynamic analyses of linked rigid-segment models. Rigid-segment models represent the body as a series of segments that are linked together at frictionless, undeformable connections (i.e. joints). Body segments are defined by their physical dimensions, mass, location of their center of mass, and moments of inertia and further rely on the assumption that the segments do not deform during locomotion. The connections between segments mean that forces and moments transfer across these joints in an equal and opposite manner. Inverse dynamics solves these biomechanical models based on measured segment masses and accelerations developed from kinematic profiles to produce net joint moments and joint reaction forces. Since motion in multibody rigid systems is complex, most inverse dynamics biomechanical models assess and correct for differences between the modeled and observed [via ground reaction forces (GRF)] motion of the whole body with residual forces and moments. Forward dynamics begins with forces (e.g. GRF) applied to the rigid segments and then solves for the resulting accelerations and assumes that optimization of parameters (such as metabolic cost) drives the motions (e.g. Sellers et al., 2005). Forward and inverse dynamic approaches are both useful and based on assumptions that guide the simulation (Demuth et al., 2023).

Musculoskeletal modeling extends rigid-segment models by including detailed models of individual muscles (which include important parameters such as pennation angle, muscle path, optimal fiber length, maximum voluntary contraction force), as well as

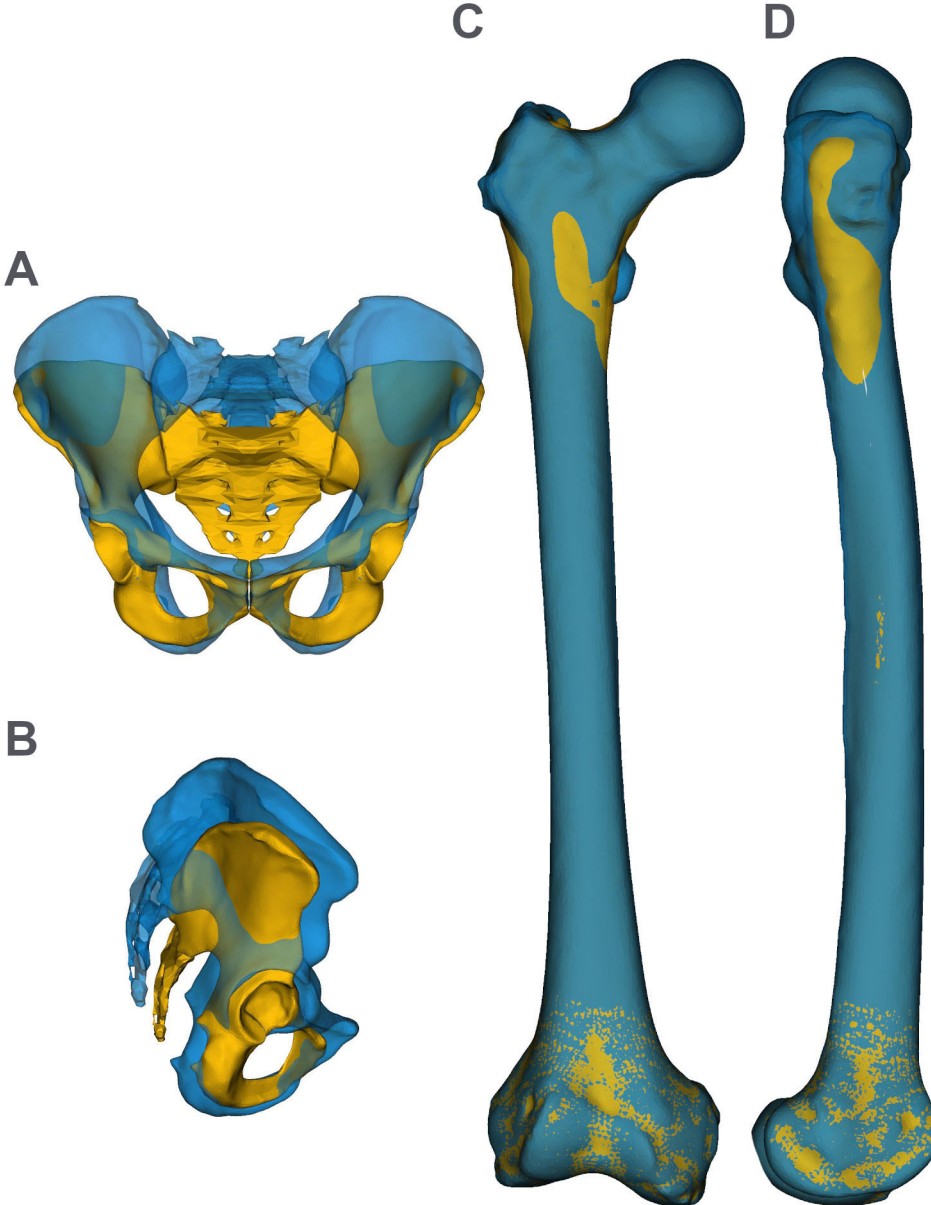

**Fig. 2. ADL modern human-like (blue) and ADL australopithecine-like (yellow) pelves in anterior (A) and lateral (B) views and femora in anterior (C) and lateral (D) views.** Note that although the acetabula are coincident, the iliac blade shape differs. The ADL australopithecine-like pelvis is compressed anteroposteriorly relative to the ADL modern human-like pelvis.

models of neurological control (e.g. when muscles are activated). In extending the inverse dynamic approach, musculoskeletal simulations solve the muscle redundancy problem (i.e. how to apportion a net joint moment to particular muscles) based on muscle strength and segment motion under a specified minimization criterion. Musculoskeletal model results reveal estimates of muscle forces and joint reaction forces induced by both the externally applied forces and muscular action.

Within paleoanthropology, Crompton and colleagues (e.g. Crompton et al., 1998; Wang et al., 2004) investigated hominin movement using musculoskeletal modeling, but their approach was necessarily limited by the lack of software available to carry out the complex calculations required to perform musculoskeletal analyses of the lower limb. The ability exists now to evaluate the impact on locomotor parameters of morphologies that differ among species (e.g. O'Neill et al., 2024). Because inverse dynamics musculoskeletal models require kinematic and kinetic inputs, modeling an extinct morphology is fraught because it is impossible to collect the data

necessary to drive the simulation. Optimization approaches, such as recent work in forward dynamics in other extinct groups (e.g. Anderson et al., 2023; Sellers et al., 2017), can produce potential solutions that might have worked in the past but introduce additional assumptions, such as what variables are important in the optimization (Demuth et al., 2023).

While some aspects of morphology can be known, forces and movements remain hypotheses. Consequently, a more tractable approach with fewer assumptions is one that modifies a modern human model to approximate the shape of an extinct form in an area of interest (e.g. the hip or foot) rather than to model the extinct form *de novo*. Our reasoning is this: we know both the human morphology and the kinematics and kinetics that humans exhibit when walking. We have information, though incomplete, about the morphology of australopithecines, but much less convincing evidence about their kinematics and kinetics. The current understanding is that the human-like morphology is derived from that of an australopithecine-like ancestor (Foley et al., 2016). A

reasonable first hypothesis is, then, that the cause of the evolution of hip morphology from australopithecine-like to the modern human-like is rooted in locomotion. We start from characteristics known to exist as a functional whole (the modern human form) and hypothesize about a change to one aspect of that system (the hip). Our approach is founded on the null hypothesis that human and australopithecine locomotion are biomechanically equivalent. We seek to reject that proposition by evaluating a set of biomechanical variables. Rejection of the null hypothesis would allow us to begin to hypothesize about the nature of differences in locomotory biomechanics and their implications. It is, of course, possible that we will not reject this hypothesis. It is also important to note that this approach – at least at this stage – does not aim to reconstruct australopithecine walking gait.

We have, therefore, modified a modern human pelvis and (proximal) femur to approximate the shape of the pelvis and femur of A.L. 288-1 as reconstructed by Lovejoy (1988) in order to assess the impact of genus-level morphological variation on the hip joint reaction forces and muscle activations. We use muscle activations, rather than muscle moment arms and/or muscle forces, in our analysis because activations are scalars that can be evaluated without regard to the muscle orientation and represent the muscle force (i.e. the higher the activation, the higher the force). Also, the body may optimize motion to limit muscle (over-)activation (McDonald et al., 2022). We hold kinematic and kinetic variables constant to explore the effect of shape in isolation from other variables. We ask: what mechanical consequences would arise if a modern human had a pelvis shaped like that of A.L. 288-1? Specifically, prior work on the subject (e.g. Crompton et al., 1998; Lovejoy et al., 1973; Ruff, 1995; Sellers et al., 2005) suggests that the hip functional complex that includes biacetabular breadth, femoral neck length, and iliac blade flare produces hip abductor muscle activations and hip joint reactions that are similar in the modern human- and australopithecine-like forms.

## RESULTS
To verify that we have achieved our intent of comparable kinematics and, hence, hip moments in the two forms after our manipulation, we compare the normalized hip moments. Fig. 3 demonstrates the agreement that we achieved (Sylvester and Kramer, 2023).

### Joint reaction forces and moments
A comparison between the two hip forms for the average hip joint normalized reaction forces for an individual is shown in Fig. 4. Hip joint forces in the direction of travel (X) are higher in the ADL modern human-like hip than the ADL australopithecine-like one, especially during the propulsive part of stance, while hip joint forces in the vertical direction (Z) are higher in the ADL australopithecine-like hip. Side (Y) joint force differences vary. These differences are statistically significant in some parts of the stride, particularly during peak propulsion, i.e. ∼50% of stride (P<0.001). Hip muscle-induced moments (data not shown but identical in magnitude to Fig. 3) are similar between forms, as expected, because they must balance the externally induced moments (Fig. 3), which are similar by design (Sylvester and Kramer, 2023).

### Muscle activations
Muscle activations are shown in Fig. 5 for the gluteal muscles, which are the focus of this investigation, and in the Supplemental Information for other anatomical muscles that cross the hip. The total muscle-induced hip moment, which is by design the same for both models, is achieved through significantly different activations

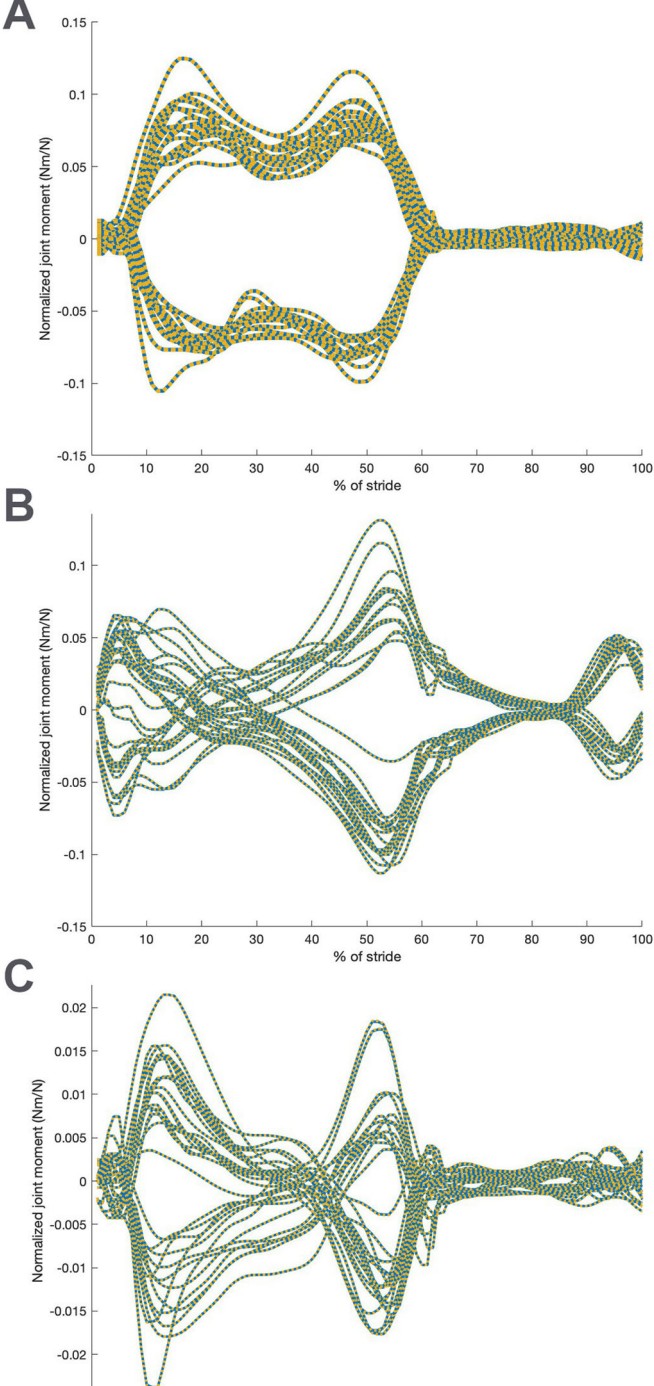

**Fig. 3. Comparison between model values and values for australopithecine-like normalized hip joint moments (Nm/N) induced by externally applied forces.** (A) Normalized moments about the model X axis (direction of travel); (B) moments about the model Y axis (mediolateral at midstance); (C) normalized moments about the model Z axis (vertical). All trials for the ADL human-like (blue) and ADL australopithecine-like (dashed gold) are shown.

in the ADL modern human and ADL australopithecine models (P<0.001) during both braking (∼15% of stride) and propulsion. Generally, the ADL australopithecine-like model requires higher

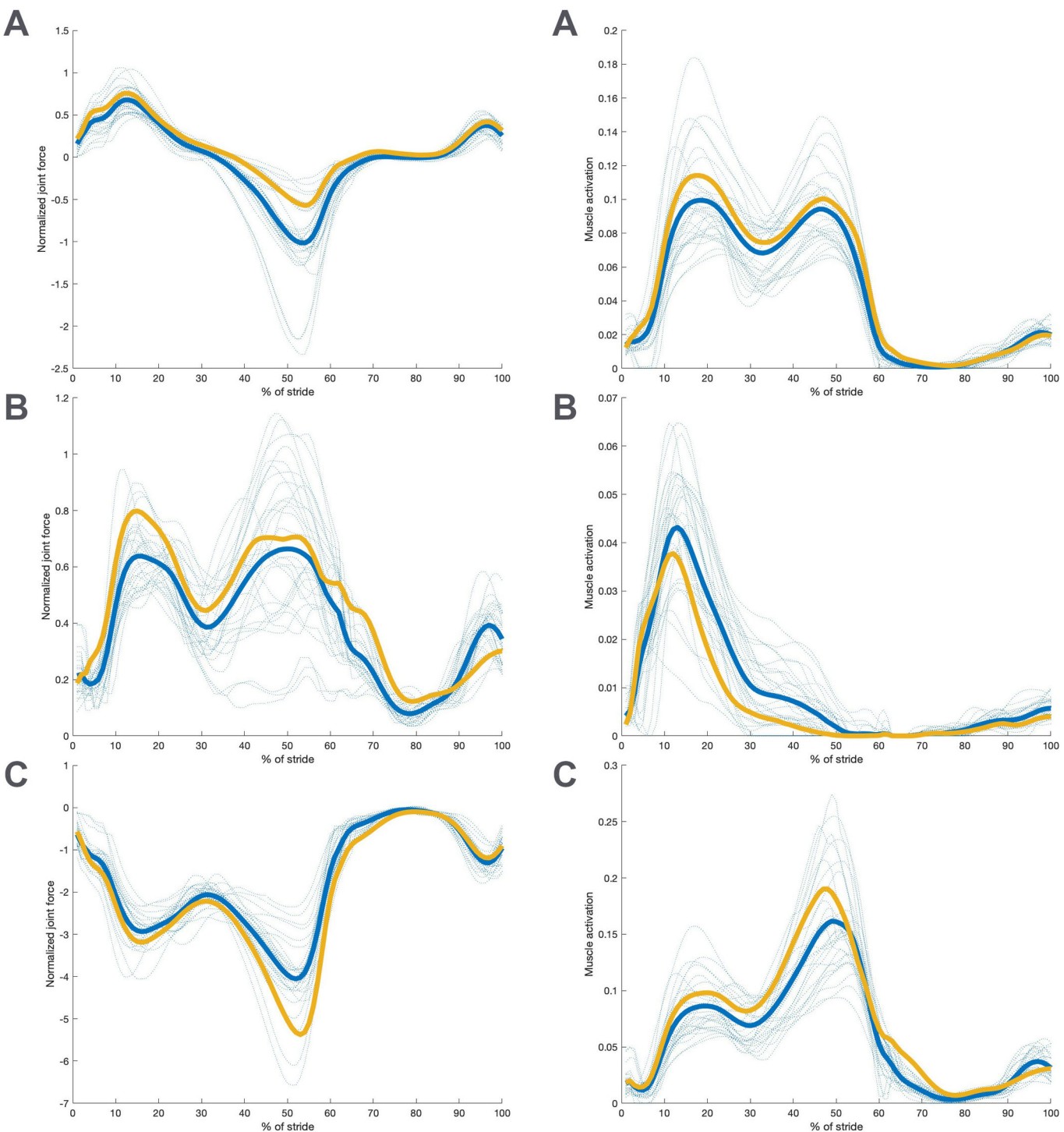

**Fig. 4. Comparison between ADL modern human-like model values and values for ADL australopithecine-like hip joint normalized reaction forces.** (A) Normalized force in the model X axis (direction of travel); (B) normalized force in the model Y axis (mediolateral at midstance); (C) normalized force in the model Z axis (vertical). All trials for the ADL human-like configuration are shown with blue dashed lines. The average for the ADL human-like (blue) and ADL australopithecine-like (gold) are shown with solid lines.

**Fig. 5. Comparison between modern ADL human-like model values and values for ADL australopithecine-like gluteal muscle activations.** (A) Activation in gluteus medius; (B) activation in gluteus maximus; (C) activation in gluteus minimus. All trials for the ADL human-like configuration are shown with blue dashed lines. The average for the ADL human-like (blue) and ADL australopithecine-like (gold) are shown with solid lines.

muscle activations in gluteus medius, but lower activations in gluteus maximus across more of the stance period than the ADL modern human-like form. Gluteus minimus is more active in the

ADL australopithecine-like hip except for several individuals that show lower activations during propulsion. Other muscles that cross the hip are generally more active in the ADL australopithecine-like model, but in most cases those muscle activations are not outside the

variation among the ADL modern human individuals (see Fig. S3). One exception to this is iliacus (see Fig. S3A).

## DISCUSSION

Our goal with this project is the first step toward understanding the walking biomechanics of australopithecine skeletal morphology. We choose to approach understanding the past by changing individual parts of the system in order to understand the impact of each. Only then can we unify the whole to understand its function. We recognize that in some cases the whole is more than the simple sum of its constituent parts and we plan, therefore, to continue to build complexity.

In this initial step, we sought to know what would happen if a modern human had an australopithecine-like hip. We started with several assumptions including that the initial simulation should utilize the kinematic profiles of humans, that scaling up an australopithecine pelvis to 'fit' the hip joint space introduced no allometric conundrums, and that the orientation of the australopithecine pelvis with regard to pelvic tilt should mimic that of a modern human pelvis. We found that the shape of the ADL australopithecine-like pelvis produces absolutely higher muscle activations in gluteus medius and gluteus minimus, but lower muscle activations across a long period of stance in gluteus maximus when kinematics and size (e.g. individual total mass and stature; segment lengths, widths, breadths; mass distribution) are held constant. Hip joint reaction forces are similarly heterogeneous: the ADL australopithecine-like hip exhibits lower forces in the direction of travel but higher forces in the vertical direction.

This indicates that achieving the required externally induced moment in the ADL australopithecine-like model requires activation of different muscles than those active in the ADL modern human-like hip. In other words, variation in skeletal form produces variation in muscle function. This also indicates that there is no one muscle activation profile in the hip that is required for hominin bipedalism. On the one hand, hip abductor muscles demonstrate generally more activation in the ADL australopithecine-like pelvis. On the other hand, activation of gluteus maximus appears to be less in the ADL australopithecine-like hip. This difference is potentially due to less favorable muscle orientations, i.e. ones that are not fully in the plane of the moment. The muscle redundancy algorithm solves for a result that minimizes muscle activation, but it works with the muscle orientations and other geometric relationships that are inherent in the particular model.

Taken together this pattern of gluteal muscle activation might indicate that the bony morphology of the ADL australopithecine-like is optimized toward hip extension capability at the expense of abduction, while the ADL modern human-like hip prioritizes abduction at the expense of hip extension. The obvious function of strong hip extension in a medium sized primate is facility in vertical climbing (Thorpe et al., 2007). With the australopithecine-like hip, terrestrial bipedalism can be easily achieved in a form also optimized for climbing. That the ADL modern human-like form tends toward a morphology that favors coronal plane pelvic stabilization over hip extension might indicate that extension is of lesser importance to modern humans than australopithecines. One possibility is that an increased distance from the ground to the body's center of mass (due to, for instance, longer legs or burdens carried on the upper body in humans) requires more stabilization from the hip abductor muscles during single stance.

This different pattern in the gluteal musculature activations between forms is also seen in the hip joint reaction forces. Joint reaction forces include contributions from both the externally

applied forces and the muscle forces, so when muscle forces vary, joint reactions do as well. The ADL australopithecine-like hip experiences higher vertical forces (Fig. 4C) but lower forces in the direction of travel, particularly during propulsion (Fig. 4A). We expect that these differences in forces might well have implications for bony morphology. Although a full discussion of the functional differences between modern human- and australopithecine-like hips are beyond the scope of this paper, and, we would argue, premature until the effect of our assumptions is evaluated in more detail in future work, we offer one example. The anterior horn of the acetabulum extends through a smaller arc of the lunate surface in the australopithecine-like acetabulum. While a diminutive anterior horn has been suggested as being a feature that indicates different kinematics (Stern and Susman, 1983), our results indicate that it might be due to the lower anteroposterior forces in the australopithecine hip.

Nonetheless, nothing in the (isolated) morphology of the hip (pelvis and proximal femur) is incongruent with modern human walking kinematics. A non-modern kinematic pattern, often referred to as bent-hip, bent-knee, has been proposed as a requirement for australopithecine bipedalism based on the position of the iliac blades in the A.L. 288-1 pelvis (Stern and Susman, 1983). Those authors claimed that the australopithecine pelvis/hip (like that of a chimpanzee) would have been unable to generate coronal plane moments because of inadequate leverage. They further argued that the moments necessary to stabilize the pelvis could only be achieved with substantial hip, and concomitant knee, flexion. In the simulations presented here, insufficient coronal plane leverage, if exhibited, could have been compensated by substantially higher muscle activations and resulting forces. The activations of lesser gluteal muscles are, however, only marginally higher in the australopithecine-like model compared to the modern human-like model demonstrating the ability of the australopithecine-like form to control coronal plane pelvic motion. While this increase might have carried higher metabolic costs, reduced gluteus maximus activations could have compensated. Our results are congruent with those of others (McDonald et al. 2022) and suggest that interpretations of non-modern walking kinematics cannot be based on pelvic morphology alone.

Although deliberate, several of our assumptions bear further discussion. First, australopithecines might have used a different kinematic profile than the modern human one that we employed. This has certainly been suggested in the past, particularly that australopithecines might have used a bent-hip-bent-knee (or more crouched position) when they walked (Stern and Susman, 1983). Usually in those analyses, the assumption has been that bent-hip-bent-knee would lead to less effective bipedalism in australopithecines and musculoskeletal evidence supports this (Carey and Crompton, 2005). Other kinematic profiles could also have been used by australopithecines, but what those might have been is difficult to know. Fossil footprint trail evidence suggests motion similar although not identical to that of modern humans (e.g. Alexander, 1984; Hatala et al., 2023). Forward dynamic simulations of australopithecine skeletal morphology (e.g. Watson et al., 2009) have produced kinematic profiles similar to those employed by modern humans – reciprocating lower limbs swing and inverted pendular motion of the center of mass. We chose, therefore, to use a kinematic pattern that is known to be evolutionarily viable. Beyond terrestrial bipedalism, the australopithecine-like form might also be selected for other typical movements, such as arboreal bipedalism and/or vertical climbing. The ability to climb well does not prevent a creature from being an effective biped (Sylvester, 2006), as known

anecdotally and as this analysis begins to demonstrate, but frequent climbing bouts might tip the selective scales toward forms with increased extension ability. These possibilities of other important movement patterns could be evaluated if appropriate kinematic and kinetic profiles could be developed.

The second assumption is that we have not oriented the australopithecine-like pelvis in the way that it was used in terrestrial bipedalism in australopithecines. We intentionally oriented the australopithecine-like pelvis as it typically is taken to be in modern humans. This choice allows us to assess the effect of shape without the confounding influence of orientation, but the orientation might have been different in australopithecines. Specifically, we might have tilted the pelvis more posteriorly than it was in life (Fig. 2). The impact of more posterior tilt than in the living creature is that the anterior hip musculature – the anterior fibers of gluteus medius, for instance – is in a less effective position for abduction stabilization and internal rotation than it is in the human situation (Zihlman and Hunter, 1972). This difference in anterior muscle fiber orientation between the two forms is due to the more anteriorly wrapped iliac blades in modern humans (Lovejoy, 1988). It is possible that anteriorly tilting the australopithecine pelvis – in other words, moving ASISs and pubic tubercles from a vertically aligned position to having the ASISs anterior to pubic tubercles – might create a situation where the abductor muscles were aligned in a more effective orientation. This might also affect muscle activation in iliacus. Although aligning the pelvis with ASIS and the pubic tubercles in a plane parallel to the anatomical coronal plane (or the global YZ plane in our AnyBody model) is standard practice, modern humans differ in their neutral position by at least ±5° (Marques et al., 2018; Mayr et al., 2005). Wiseman (2023) and Bates et al. (2025) chose what we consider to be an improbable posterior tilt that produces a near horizontal sacral slope. Future work will explore the consequences for muscle activations of varying amounts of pelvic tilt in modern humans and australopithecines.

The final assumption involves the question of allometry. We assume in this analysis that the australopithecine pelvis – a morphology that existed in creatures that were considerably smaller, both in terms of linear dimensions and mass than most modern humans – could be scaled isometrically to the width of a modern human pelvis. Evidence suggests that this assumption of isometry is incorrect and could have important implications for the analysis we present here (e.g. Jungers, 1991). We explicitly used the pelvis of A.L. 288-1, a small australopithecine only slightly larger than Sts 14, because A.L. 288-1 has a proximal femur, an innominate, and a sacrum (although with taphonomic damage that must be reconstructed). Some ancient hominins [e.g. *Ardipithecus ramidus* (White et al., 2009) and KSD-V/P-1-1 Kadanuumuu (Haile-Selassie et al., 2010)] are, however, of a mass and stature within the modern human range, but their pelvic morphology is less understood than that of the smaller-bodied australopithecines. Later forms such as the Gona pelvis are shaped differently from both australopithecine-like and modern human-like pelves (Simpson et al., 2008) and would, presumably, exhibit different muscle activations than either. Intriguingly, *Homo floresiensis* (LB1) is similar to A.L. 288-1 and Sts 14 in both size and pelvic shape (Brown et al., 2004; Jungers et al., 2009), supporting the need to understand the impact of size on morphology. Consequently, the question of scaling and size remains critical and will be addressed in future research.

## Conclusion

The shape of an australopithecine pelvis in a modern human body produces higher hip muscle activations for gluteus medius, but lower activations for gluteus maximus, when evaluated with our assumptions about kinematics, kinetics, scaling, and pelvic orientation. Similarly, vertical hip joint forces are higher in the australopithecine-like form while forces in the direction of travel are less. Given that creatures are adapted for the movement profiles typical of their daily activities, the results for the australopithecine-like hip suggests that, in addition to effective terrestrial bipedalism, activities that required hip extension were also important to australopithecines. These tantalizing initial results speak strongly for the need to approach understanding the locomotion of extinct creatures from a foundation grounded in biomechanics and with the perspective that walking unburdened in a straight terrestrial path on even and level terrain is but the tip of the iceberg of biomechanical analyses that must be undertaken to explicate the functional evolution of this lineage. The fossil record and our initial analysis makes clear that small-bodied australopithecine hip morphology is its own evolutionarily stable form – adapted by the selective pressures imposed from the environmental conditions of their time in the context of their evolutionary history – as is, presumably, that of modern humans.

## MATERIALS AND METHODS
### MoCap
We utilize the self-selected walking (C4) trials of Schreiber and Moissenet (2019) as the baseline human configuration using an approach described elsewhere (Kramer and Sylvester, 2023). We selected five female and five male subjects to represent the breadth of size (stature and body mass) represented in the original sample and analyzed 45 trials (Table 1). The dataset includes motion capture data (i.e. the motion of 52 markers placed on anatomical landmarks), ground reaction forces from two force plates embedded in the floor, gait events assessed by Schreiber and Moissenet (2019), and anthropometric values. These walking trials produce typical human walking kinematic patterns and muscle forces (see Figs S1 and S2). We use these data to drive simulations of the modern human- and australopithecine-like (described below) musculoskeletal models (Fig. 1). Our comparisons are between the original and modified models, i.e. the original configuration acts as its own control to examine the model's response to modification of hip shape.

These data are used to drive the analysis of a well-established musculoskeletal model (the ADL_Gait [beta] full body MoCap human model AnyBody Version 7.3.4) of a modern human that has nine rigid segments (pelvis, left and right thigh, shank, talus, foot) and 169 muscle elements in the pelvis and lower limb using AnyBody (v7.3, AnyBody Technology, Denmark) and well as segments for the head, arms, and trunk (Fig. 1). In order to emphasize that we start from a model derived from an individual, we refer to this model as the 'ADL modern human' and segments of the model (e.g. pelvis) use this convention (e.g. ADL human pelvis). The lower limbs of this model are based on the TLEM 2.1 validated lower limb model (Carbone et al., 2015; De Pieri et al., 2018). To simulate the muscle activations and joint reaction forces, the generic version of the model is first scaled to the size (body mass and stature) of the individual from whom the motion data were collected, and then the internal proportions of the individual are optimized. Because some joint locations are impossible to determine without invasive imaging (e.g. the hip joint), the software performs an optimization analysis to determine the set of joint locations that best fit the motions. This step is called parameterization (or in AnyBody ParameterIdentification), and we use dynamic trials for parameter optimization.

A central issue in creating a parameterized model is optimizing the pairing of the experimental marker data that encapsulate the movement of the moving humans with the virtual markers that act as drivers of the musculoskeletal model. The parameterization algorithm can accommodate optimization of most marker pairings (e.g. markers on the greater trochanter or lateral knee), but it requires some pairings to be linked (i.e. the virtual markers that drive the model are forced to the position of the experimental markers from the motion of the individual) in order to ground the model.

Additionally, establishing the neutral position of the pelvis segment requires more constraint for the algorithm to achieve anatomical pelvic position than other segments. We accept an anterior pelvic plane (defined by the anterior superior iliac spines and the pubic tubercles) within ±5° of parallel to the coronal plane as neutral (Marques et al., 2018; Mayr et al., 2005). The experimental and virtual markers that determine pelvic alignment, or tilt, in the sagittal plane are different from those that determine the anterior pelvic plane: both include the anterior superior iliac spines, but the model uses the posterior superior iliac spines rather than the pubic tubercles to define the anterior pelvic plane. Because variation in pelvic bony morphology can vary the approximation of the anterior pelvic plane by >20° (Preece et al., 2008), the location of the pelvic markers must be determined for each individual. We aimed to achieve ±5° from vertical in the standing model and an average across the gait cycle of ±5° (Brown-Taylor et al., 2020).

The product of the parameterization process is a model that has overall (mass and stature) and segment (widths and lengths) sizes specific to the individual, but generic segment shapes. Once the generic model has been parameterized, the marker motions and ground reaction forces are used to drive the inverse dynamic simulation and the solution of the muscle redundancy problem. The results include muscle element activations and joint reaction forces.

### Morphing

To produce an ADL australopithecine-like hip morphology, we modified the generic pelvis (innominates and sacrum) and proximal femur to the australopithecine shape by utilizing the ability of AnyBody to transform the shape of a segment from the generic to a 'subject-specific' one. Here our subject is based on the reconstruction of A.L. 288-1 accomplished by Lovejoy (1988). Other reconstructions of A.L. 288-1 are available (Brassey et al., 2018; Häusler and Schmid, 1995) as are those of other australopithecines (e.g. Claxton et al., 2016; Häusler and Schmid, 1995). We chose to focus on A.L. 288-1 because it includes an associated femur. The Brassey reconstruction of A.L. 288-1 (Brassey et al., 2018), which was developed to determine segment masses, and the adjustment made by Wiseman (Wiseman, 2023) do not reconstruct the damage noted in the original fossil description (Johanson et al., 1982) to the auricular and retroauricular regions. Both the Lovejoy reconstruction and the Häusler and Schmid reconstruction do correct this damage. The Lovejoy version of A.L. 288-1 (Lovejoy, 1988) is 2 mm narrower in the anteroposterior and 3 mm broader in the mediolateral dimensions than the Häusler and Schmid reconstruction (Häusler and Schmid, 1995), making the Lovejoy reconstruction more oval than that of Häusler and Schmid. That is, the Häusler and Schmid version is closer to a birth canal shape of a modern human (Häusler and Schmid, 1995). We suspect that in the context of gait mechanics the modest metric differences between the Lovejoy reconstruction and that of Häusler and Schmid would have little impact on muscle activations during walking relative to the modern human condition. We selected the Lovejoy reconstruction because of chronological precedence and widespread availability.

Presented in full elsewhere (Sylvester and Kramer, 2023), we briefly describe here our procedure to morph the generic ADL pelvis and proximal femur models to approximate that of the Lovejoy pelvis reconstruction and the A.L. 288-1 proximal femur (A.L. 288-1ap). For the pelvis, we began by reducing the asymmetry present in Lovejoy reconstructed A.L. 288-1 pelvis (Sylvester and Kramer, 2023) following the procedures outlined by Gunz et al. (2009). Next, we identified 63 landmarks on both pelves and calculated the thin-plate spline (TPS) interpolation function that mapped the AnyBody pelvic landmarks to the exact location of the A.L. 288-1 pelvic landmarks. We then applied the parameterized TPS function to the vertices of the ADL pelvis surface model. This generated a pelvis with the shape of the A.L. 288-1 Lovejoy reconstruction but with the surface model triangulation of the AnyBody pelvis model (i.e. identical number of vertices and connectivity). This facilitated morphing the pelvis within the AnyBody modeling software (which is described below).

To complete morphing of ADL australopithecine hip, we also created a hybrid femur that included a proximal end that reflected the A.L. 288-1 femur and a distal end that retained the morphology of the ADL femur (as described in detail in Sylvester and Kramer, 2023). We first scaled A.L.

288-1 femur to have length of the AnyBody femur surface model and then surface aligned the proximal 20% using the iterative closest point algorithm (Besl and McKay, 1992). Next, we evenly distributed 237 vertices (Cartesian coordinates) on the proximal 20% of AnyBody femur surface model using the Gibbon package for Matlab (Moerman, 2018). We established geometrically homologous locations on A.L. 288-1 femur using the coherent point drift algorithm (λ=1; staged β=50, 30, 10, 8) (Myronenko and Song, 2010). For the distal end, we identified 2500 vertices of the ADL femur surface model. From these three sets of points, we created two landmark sets: the first represented the human ADL femur: human proximal and distal vertices; and second represented the hybrid femur: australopithecine proximal and human distal vertices. We calculated the TPS interpolation function that mapped the human landmarks to the hybrid femur and then applied the function to the ADL femur vertices. The result is a femur surface model that has a shaft that is a smooth, TPS-based interpolation between a proximal end shaped by australopithecine morphology and a distal end which reflects the ADL human femur.

### Warping ADL in AnyBody system

The AnyBody modeling software has the capability to deform skeletal elements through various functions. In our study, we opted for the TPS-based warping function, which employs landmarks to govern the deformation, similar to the previously mentioned TPS methods. Given that the ADL australopithecine pelvis and femur surface models share the same triangulation as their human counterparts, the process only necessitated the identification of an adequate number of vertices on the surface models. Specifically, we pinpointed 145 vertices on the pelvis and 604 on the femur to serve as landmarks for the TPS-based morphing in AnyBody. These AnyBody scripts are available in the Supplementary Information. In the case of the pelvis, these landmarks encompassed not only points on its surface but also included the left and right hip centers, along with the midpoint of the two anterior superior iliac spines. As for the femur, 101 vertices represented the axis of knee flexion-extension through linear interpolation between defined joint axis points. Additionally, one vertex denoted the center of the hip in the femoral head, while the remaining vertices represented the bone's surface. To execute the morphing process within the AnyBody software, we developed appropriate script files that were implemented during the model loading phase.

### Scaling

To preserve the integrity of the parameterization (described above), we maintained the hip joint locations which preserves the hip breadth that was determined for each human subject. That is, while the pelvis and proximal femur were morphed to reflect the shape of australopithecine-like morphology, these elements were scaled to have distances between joint centers (i.e. hip to hip distance, hip to knee distance) that were identical to the ADL modern human-like model. Finally, relative to the ADL modern human-like model, we did not change the lumbar spine and structures superior to it nor the location of the knee and morphology distal to it. The TPS-based morphing described above shifts the hip musculature path geometry as appropriate to the change in shape of the bones. We checked this modified geometry for malalignment or interference with muscle path geometry. The final product of this transformation is a pelvis and proximal femur with australopithecine-like shape but the size of the original human participant.

Because the segment size is adjusted to match the kinematic information (obtained from the locations of the experimental markers determined in the motion trials) of the ADL modern human, we developed a pelvis and paired femora that maintained the size of ADL modern human [i.e. the same pelvic breadth (distance between hip joint center) and femoral length (distance between hip and knee joints)] but changed the shape to match that of A.L. 288-1. That is, we have a model that is the size and internal proportions of each of the human participants but with the hip morphology of an australopithecine. This approach maintains the muscle origin and insertion locations for the australopithecine form in the same position relative to the skeletal element as in the human shape.

After the pelvis was morphed, it was aligned in the sagittal plane. (The orientations in the coronal and horizontal planes are established by the

requirement of left-right symmetry.) The rule-of-thumb approach that is used for modern human pelves of aligning the anterior superior iliac spines and pubic tubercles to produce a vertical orientation is not necessarily appropriate for nonhuman pelves (Kozma et al., 2018; Lovejoy, 2005). The maximum range of tilt in a bipedal pelvis is determined by the requirement for sufficient fibers of gluteus medius and minimus to stabilize the pelvis from pelvic drop during the single stance phase of walking, but the extremes would produce lumbar curves, pelvic tilts, and/or sacral slopes that are not feasible (figure 3 in Kozma et al., 2018). Consequently, we have rotated the morphed pelvis to the orientation of the human pelvis, which achieves a sacral slope of ∼30° (Lovejoy, 2005) and a lumbar curvature of ∼40°. This lumbar curvature is in the lower range predicted by Been et al. (2014). This orientation of the morphed pelvis aligns the iliac blade and, hence, the origins of gluteus medius and minimus superior to the acetabulum (Fig. 2) similar to figure 9 in Lovejoy (2005) and in the mid-range of Kozma et al. (2018) (see figure 3). We note that this is not the alignment chosen by Wiseman (2023) or Bates et al. (2025) in their models, which we estimate produced a sacral slope of ∼18° and a lumbar curvature of <10°.

### Analysis (comparison of the ADL modern human form to the ADL australopithecine-like form with same kinetics and kinematics)

Our goal is to compare the effect of the shape difference between an australopithecine-like and a modern human-like pelvis on hip joint reaction forces and hip muscle activations. Muscle activations represent the degree to which a muscle develops force relative to its maximum strength (or maximum voluntary contraction value), which is an input characteristic of each muscle in the model. An activation of 0.1 indicates that the muscle element experiences a force of 10% of its maximum strength. The force that is apportioned to a muscle is determined by segment motion using a direct solution of the dynamic equilibrium equations and a given muscle's instantaneous strength, in an algorithm that minimizes the sum of activations raised to an exponent (here we use an exponent of 3) (Andersen, 2021). Morphing the shape of the skeletal elements as is done in this work does not affect muscle parameters that are determined by body mass and the relevant segment dimensions. Activations are scalars and can, therefore, be mathematically manipulated (e.g. summed, averaged) without consideration of their orientation in space. In addition, they represent the state of the muscle relative to its maximum capacity.

Individual average curves are calculated for hip joint reaction forces for the ADL modern human- and ADL australopithecine-like hips. Similar to previous work (Sylvester and Kramer, 2023), we aggregate the activations and forces of the muscle elements into recognized anatomical muscles (e.g. the 12 elements of gluteus medius are combined). The ADL modern human-like trials are a subset from Kramer and Sylvester (2023) and produce muscle activations that are similar to Sylvester et al. (2021) (Supplemental Information). To normalize force and moment data, we divide the magnitude produced through the simulation by the individual's body weight. Individual average activation and normalized force curves for those anatomical muscles are also calculated.

The original data include ground reaction forces from two force plates. A complete stride from initial contact with the first force plate to ipsilateral initial contact (assessed via visual inspection of the optical markers by Schreiber and Moissenet, 2019) is analyzed. We convert the Y axis (mediolateral at midstance) joint reaction forces of left feet to pseudo-right feet by changing their sign. This allows us to compare the Y axis joint reaction forces directly. Of note, muscle element moment arms are not determined in the inverse dynamics solution implemented in AnyBody but can be calculated from muscle velocity and joint angular velocity (An et al., 1984). We provide the moment arms for idealized joint rotations in the Supplemental Information to facilitate comparison to static or anatomically based studies.

We use statistical parameter mapping (SPM), a technique that allows for the evaluation of dependent variables that vary across a cycle and is particularly useful to assess gait variables that vary with time (Pataky et al., 2013, 2014), to assess the magnitude and the shape of the curves. Analysis of variance (ANOVA) with one factor (modern human-like versus australopithecine-like hip) and repeated measures (three kinematic/kinetic trials for each subject) were implemented in Matlab through the spm1d

package (spm1d.org). SPM ANOVA requires a balanced design, so we include in the analysis three walking trials from each participant in the statistical analysis (i.e. three trials from each of ten participants for each hip configuration).

### Acknowledgements
We thank the editors and two anonymous reviewers for their careful and thorough review of original versions of this paper.

### Competing interests
The authors declare no competing or financial interests.

### Author contributions
Conceptualization: P.A.K., A.D.S.; Data curation: P.A.K., A.D.S.; Formal analysis: P.A.K., A.D.S.; Investigation: A.D.S.; Methodology: P.A.K., A.D.S.; Project administration: P.A.K., A.D.S.; Resources: P.A.K., A.D.S.; Software: P.A.K., A.D.S.; Validation: P.A.K., A.D.S.; Visualization: P.A.K., A.D.S.; Writing – original draft: P.A.K., A.D.S.; Writing – review & editing: P.A.K., A.D.S.

### Funding
 Deposited in PMC for immediate release.

### Data and resource availability
All relevant data can be found within the article and its supplementary information. Full information of the musculoskeletal model used in this study can be found here: https://anyscript.org/ammr/Applications/Mocap/ADL_Gait.html. C3D files for motion capture data from Schreiber and Moissenet (2019) that were used in the study can be found here: https://figshare.com/articles/dataset/A_multimodal_dataset_of_human_gait_at_different_walking_speeds/7734767. Additional data that support the findings of this study are available from the corresponding author upon reasonable request.

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
