## [Peer Review File · Biology Open]

Hip stabilization in an australopithecine-like hip: the influence of shape on muscle activation

Adam David Sylvester and Patricia Ann Kramer

DOI: 10.1242/bio.061931

Editor: Lewis Halsey

Review timeline

Original submission: 19 February 2025

Editorial decision: 3 April 2025

First revision received: 19 May 2025

Accepted: 20 May 2025

Original submission

First decision letter

MS ID#: bio.061931

MS TITLE: Hip stabilization in an australopithecine-like hip: the influence of shape on muscle activation

AUTHORS: Adam David Sylvester and Patricia Ann Kramer

I have now reached a decision on the above manuscript.

The reviewer reports are shown at the bottom of this email or can be accessed, together with a copy of this decision letter, by going to:

As you will see, the reviewers raised a number of substantial criticisms that prevent me from accepting the paper at this stage.

They suggest, however, that a revised version might prove acceptable, if you can address their concerns. If you think that you can deal satisfactorily with the criticisms on revision, I would be pleased to see a revised manuscript. We would then return it to the reviewers.

At this stage, we also ask you to ensure your manuscript complies with our formatting guidelines. Provided you are able to fully address the referees' comments, we are positive about publication of your paper (we accept over 95% of revision submissions) and therefore hope you won't mind any extra work involved in reformatting your manuscript at this point.

Please ensure that you clearly highlight all changes made in the revised manuscript. Please avoid using 'Tracked changes' in Word files as these are lost in PDF conversion.

I should be grateful if you would also provide a point-by-point response detailing how you have dealt with the points raised by the reviewers in the 'Response to Reviewers' box. Please attend to all of the reviewers' comments. If you do not agree with any of their criticisms or suggestions please explain clearly why this is so.

Reviewer 1

Comments for the author

The article titled "Hip stabilization in an australopithecine-like hip: the influence of shape on muscle activation" investigates the impact of australopithecine-like hip morphology on muscle activation and hip joint reaction forces. The authors compare musculoskeletal models of modern human and australopithecine-like hips to explore how changes in hip morphology affect muscle activation patterns and joint reaction forces. This study provides preliminary insights into the gait and evolution of australopithecines, but there are some shortcomings in the paper that could affect the overall quality of the research. The following are the comments from the reviewer:

1. Ensure that the terms related to hip morphology, such as "australopithecine-like pelvis" and "modern human-like pelvis," are clearly defined early in the paper for readers unfamiliar with the specifics of these skeletal variations.
2. The article assumes modern human kinematics as a baseline. It would be beneficial to explore alternative kinematic models or clarify the limitations of this approach, especially considering the extinct nature of australopithecine locomotion.
3. The muscle activation algorithm could be better explained for readers who are unfamiliar with the specific mechanics used to calculate these activations, particularly the muscle redundancy algorithm.
4. The study focuses on comparisons between the australopithecine-like and human-like pelvis models. It might be useful to also include comparisons with other hominin species, such as *Paranthropus* or *Homo erectus*, to provide a broader evolutionary context.
5. Discuss the potential biases introduced by assuming modern human kinematics and scaling extinct forms to human sizes. These assumptions could limit the accuracy of the results and should be addressed more thoroughly in the discussion.
6. The manuscript would benefit from clearer visual representations of the models used (e.g., figures depicting the pelvis and femur morphing processes, and muscle activation patterns). Adding annotated figures could help readers understand complex anatomical changes.
7. Expand the discussion on how the differences in muscle activation and joint reaction forces between the two pelvic forms impact our understanding of bipedal evolution, particularly concerning climbing and walking capabilities.
8. The orientation of the pelvis in the models follows modern human standards. It would be interesting to test other plausible orientations for the australopithecine pelvis, which may offer different insights into their bipedal capabilities.
9. Acknowledge the limitations of musculoskeletal modeling, particularly in simulating extinct morphologies, and how these limitations could impact the study's findings. The research by D. Xu, et al. on musculoskeletal modeling in motion analysis may be of significant help to the authors. It is recommended that the authors refer to this work. (New insights optimize landing strategies to reduce lower limb injury risk (<https://doi.org/10.34133/cbsystems.0126>))
10. The paper could further elaborate on the functional consequences of muscle activation differences, particularly concerning how these might have influenced other aspects of hominin behavior, such as tool use, carrying, or long-distance walking.

Reviewer 2*Comments for the author*

This paper builds incrementally on previous works by the authors on reconstructing Australopithecine anatomy in simulation environments. Here, they focused on the effects of morphing a model of human pelvic anatomy into an australopithecine form on joint reaction forces. I like the core idea of 'what happens if we scale up an australopithecine pelvis to the size of a modern human-how does it function?'

While the main findings of the paper are adequately described, there are issues with engagement with the relevant literature and reproducibility of the results that need addressing before this manuscript is suitable for publication in *Biology Open*.

Engagement with the literature.

Firstly, the engagement with the literature on australopithecine pelvic anatomy is very brief and requires more in-depth engagement. The use of Lovejoy's reconstruction of AL288-1 needs more

justification, as the 'platypeloid' shape proposed by Lovejoy is highly controversial within palaeoanthropology. It is likely that this extreme reconstruction will have significant influence on the joint reaction forces reported here. There is no reference to alternative reconstructions such as Haeusler and Schmidt's (1995) or Brassey et al.'s (2018), both of which resulted in larger dimensions.

The sacral angle used may have some issues. Firstly, there is disagreement in the literature as to how this is measured, and the authors need to be clear as to how their angle is derived. The recent reconstruction by Wiseman (2023) is critiqued for an unrealistic sacral angle (which this reviewer agrees with) but neglects to mention that this is a modified version of the Brassey et al.'s 2018 reconstruction, which is freely available. Although Brassey et al. (2018), did not report the sacral angle, it would be simple for the authors to measure and report this (even if they disagree with the reconstruction). The article cited in text by the authors does not appear to actually have reported the sacral angle, so this citation may require revision. An in-depth review of os coxa morphology in australopithecines which may assist with further clarification is that by Crompton et al. (2024). The literature review of multibody dynamic simulation in studies of extinct animals needs more reference to more recent work and reviews (e.g. Demuth et al., 2023, Brassey et al., 2018, Bates et al., 2025).

Page 4, line 3-23 needs revision to take account recent work by Wiseman (2023) and Bates et al. (2025).

Reproducibility of analyses.

All transformations of models are described in very general terms and the readers are referred to scripts which are not provided for exact methodologies. The underlying 3d models analysed in this MS are not provided-this is necessary for people to be able to properly evaluate the findings, especially when submitting to a Journal titled 'Biology Open' and would be in line with best practice in the field.

Other points

Page 12 lines 12-19. This is unclearly written and needs revising.

Page 12 lines 37-44. I was under the impression that certain aspects of Australopithecine pelves have muscle attachment sites which differ considerably in volume to modern humans (even if scaled up) and that the position of these differed somewhat. This means that the model may superficially look like an australopith's but doesn't really bear a biological resemblance.

References

- Bates, K. T., McCormack, S., Donald, E., Coatham, S., Brassey, C. A., Charles, J., ... & Sellers, W. I. (2025). Running performance in *Australopithecus afarensis*. *Current Biology*, 35(1), 224-230.
- Brassey, C. A., Maidment, S. C. and Barrett, P. M. (2017). Muscle moment arm analyses applied to vertebrate paleontology: a case study using *Stegosaurus stenops* Marsh, 1887. *J. Vertebr. Paleontol.* 37, e1361432. <https://doi.org/10.1080/02724634.2017.1361432>
- Brassey, C. A., O'Mahoney, T. G., Chamberlain, A. T., & Sellers, W. I. (2018). A volumetric technique for fossil body mass estimation applied to *Australopithecus afarensis*. *Journal of Human Evolution*, 115, 47-64.
- Crompton, R., Elton, S., Heaton, J., Pickering, T., Carlson, K., Jashashvili, T., ... & McClymont, J. (2024). Bipedalism or bipedalisms: The os coxae of StW 573. *Journal of Anatomy*.
- Demuth, O. E., Herbst, E., Polet, D. T., Wiseman, A. L., & Hutchinson, J. R. (2023). Modern three-dimensional digital methods for studying locomotor biomechanics in tetrapods. *Journal of Experimental Biology*, 226(Suppl_1), jeb245132.
- Häusler, M., & Schmid, P. (1995). Comparison of the pelves of Sts 14 and AL288-1: implications for birth and sexual dimorphism in australopithecines. *Journal of Human Evolution*, 29(4), 363-383.
- Wiseman, A. L. (2023). Three-dimensional volumetric muscle reconstruction of the *Australopithecus afarensis* pelvis and limb, with estimations of limb leverage. *Royal Society Open Science*, 10(6), 230356.

Reviewer's Responses to Questions

Experimental quality

Does each figure have the proper controls?

If 'No', please indicate reasons in Comments for Author box below.

Reviewer #1:

No

Reviewer #2:

Yes

Were the data analyzed using appropriate statistical tests?
If 'No', please indicate reasons in Comments for Author box below.

Reviewer #1:

Yes

Reviewer #2:

Yes

Reproducibility

Were experiments performed using adequate number of biological replicates?
If 'No', please indicate reasons in Comments for Author box below.

Reviewer #1:

Yes

Reviewer #2:

No

Does the methods section provide sufficient detail to permit reproducibility?
If 'No', please indicate reasons in Comments for Author box below.

Reviewer #1:

No

Reviewer #2:

No

Completeness

Are the manuscript's conclusions supported by the data?
If 'No', please indicate reasons in Comments for Author box below.

Reviewer #1:

No

Reviewer #2:

Yes

Scholarship

Do the authors cite and discuss the merits of data that would argue for and against their conclusion?
If 'No', please indicate reasons in Comments for Author box below.

Reviewer #1:

Yes

Reviewer #2:

No

First revision

Author response to reviewers' comments

Comments from the Reviewers:

Reviewer 1: The article titled "Hip stabilization in an australopithecine-like hip: the influence of shape on muscle activation" investigates the impact of australopithecine-like hip morphology on muscle activation and hip joint reaction forces. The authors compare musculoskeletal models of modern human and australopithecine-like hips to explore how changes in hip morphology affect muscle activation patterns and joint reaction forces. This study provides preliminary insights into the gait and evolution of australopithecines, but there are some shortcomings in the paper that could affect the overall quality of the research.

The following are the comments from the reviewer:

1. Ensure that the terms related to hip morphology, such as "australopithecine-like pelvis" and "modern human-like pelvis," are clearly defined early in the paper for readers unfamiliar with the specifics of these skeletal variations.

Thank you for this suggestion. We have added explicit information before we use the terms: "...The differences in morphology between australopithecines (we refer to this morphology as "australopithecine-like"), of which A.L. 288-1 provides a mostly complete pelvis) and modern humans ("modern human-like") can be categorized ..."

2. The article assumes modern human kinematics as a baseline. It would be beneficial to explore alternative kinematic models or clarify the limitations of this approach, especially considering the extinct nature of australopithecine locomotion.

The question that we seek to address is the impact of shape on muscle forces given modern human kinematics. Our approach is to start from model inputs that we can observe and those are limited to human data and fossil morphology. Exploring other kinematic profiles of habitual primate bipeds--if such were available-- could be interesting, but would be beyond the scope of this work because it is an entirely different question than the one we explore herein. That is, using human kinematics and kinetics is not a limitation of our work, but rather a deliberate experimental choice.

3. The muscle activation algorithm could be better explained for readers who are unfamiliar with the specific mechanics used to calculate these activations, particularly the muscle redundancy algorithm.

Changed to: "... Muscle activations represent the degree to which a muscle develops force relative to its maximum strength (or maximum voluntary contraction value), which is an input characteristic of each muscle in the model. An activation of 0.1 indicates that the muscle element experiences a force of 10% of its strength. The force that is apportioned to a muscle is determined by segment motion using a direct solution of the dynamic equilibrium equations and the a given muscle's instantaneous strength, in an algorithm that minimizes the sum of activations raised to an exponent (here we use an exponent of 3) [66]."

4. The study focuses on comparisons between the australopithecine-like and human-like pelvis models. It might be useful to also include comparisons with other hominin species, such as *Paranthropus* or *Homo erectus*, to provide a broader evolutionary context.

This might be an interesting extension of this work. Indeed, we have created models with a pelvis shaped like the Kebara 2 individual. There are two reasons for only using the australopithecine-like condition in this work: 1) australopithecines are as far back in evolutionary time as we can currently go from modern humans. Although the earliest hominin may have lived 6 million years ago, we do not have reliable fossil pelvises and femora from older species. Consequently, modern human-like and australopithecine-like shapes bracket the known data. 2) Our approach, that is this study, has not yet been vetted through publication. Once the approach is established, other morphologies and anatomical regions will be explored.

5. Discuss the potential biases introduced by assuming modern human kinematics and scaling extinct forms to human sizes. These assumptions could limit the accuracy of the results and should be addressed more thoroughly in the discussion.

We reiterate that we are not trying to recreate australopithecine locomotion. Our study is constructed as a “what if” not a “how did” scenario. The use of modern human kinematics (and hence scaling) is inherent in the development of the approach. To make this clearer, we added: “Other kinematic profiles could also have been used by australopithecines, but what those might have been is difficult to know. Fossil footprint trail evidence suggests motion similar although not identical to that of modern humans (e.g. [73,74]). Forward dynamic simulations of australopithecine skeletal morphology (e.g. [75]) have produced kinematic profiles similar to those employed by modern humans—reciprocating lower limbs swing and inverted pendular motion of the center of mass. We chose, therefore, to use a kinematic pattern that is known to be evolutionarily viable. ...”

6. The manuscript would benefit from clearer visual representations of the models used (e.g., figures depicting the pelvis and femur morphing processes, and muscle activation patterns). Adding annotated figures could help readers understand complex anatomical changes.

We agree that these details are important which is why we validated the morphing portions of this project in an initial paper that is freely available (Sylvester and Kramer 2023). We could recreate the complex figures that visually describe the process in that paper but that seems redundant.

7. Expand the discussion on how the differences in muscle activation and joint reaction forces between the two pelvic forms impact our understanding of bipedal evolution, particularly concerning climbing and walking capabilities.

We were loath to over-interpret our results because this is but the first step in what we see as a series of careful experimental manipulations of skeletal form. Nonetheless, given the encouragement from the reviewer, we have added:

“Nonetheless, nothing in the (isolated) morphology of the hip (pelvis and proximal femur) is incongruent with modern human walking kinematics. A non-modern kinematic pattern, often referred to as bent-hip, bent-knee, has been proposed as a requirement for australopithecine bipedalism based on the position of the iliac blades in the A.L. 288-1 pelvis [24]. Those authors claimed that the australopithecine pelvis / hip (like that of a chimpanzee) would have been unable to generate coronal plane moments because of inadequate leverage. They further argued that the moments necessary to stabilize pelvis could only be achieved with substantial hip, and concomitant knee, flexion. In the simulations presented here, insufficient coronal plane leverage, if exhibited, could have been compensated by substantially higher muscle activations and resulting forces. The activations of lesser gluteal muscles are, however, only marginally higher in the australopithecine-like model compared to the modern human-like model demonstrating the ability of the australopithecine-like form to control coronal plane pelvic motion. While this increase might have carried higher metabolic costs, reduced gluteus maximus activations could have compensated. Our results are congruent with those of others [e.g. 43] and suggest that interpretations of non-modern walking kinematics cannot be based on pelvic morphology alone.”

8. The orientation of the pelvis in the models follows modern human standards. It would be interesting to test other plausible orientations for the australopithecine pelvis, which may offer different insights into their bipedal capabilities.

We agree that exploring the impact of pelvic orientation is an interesting question. We are currently evaluating this question in modern humans and our preliminary results indicate that human neutral (or close to it) produces the lowest muscle activations, which is unsurprising. We agree that examining the impact of pelvic tilt in an australopithecine-like hip is a logical next step that might yield insight into habitual posture in extinct creatures. If the approach that we document here is accepted, we would begin exploring this question.

9. Acknowledge the limitations of musculoskeletal modeling, particularly in simulating extinct morphologies, and how these limitations could impact the study's findings. The research by D. Xu, et al. on musculoskeletal modeling in motion analysis may be of significant help to the authors. It is recommended that the authors refer to this work. (New insights optimize landing strategies to reduce lower limb injury risk (<https://doi.org/10.34133/cbsystems.0126>))

Thank you for the suggestion. There is no doubt that models are not reality and many assumptions must be made, but this is true for models of living organisms as well as extinct ones. We don't know, for instance, the material properties of ligaments or the influence of the many assumptions built into models of humans (Xu et al 2024). In this paper, we attempt to control for the many assumptions by making them the same between forms. We reiterate, though, that we are not attempting to recreate an australopithecine, but rather understand what it would mean for a modern human to have an australopithecine-like shaped hip.

10. The paper could further elaborate on the functional consequences of muscle activation differences, particularly concerning how these might have influenced other aspects of hominin behavior, such as tool use, carrying, or long-distance walking.

Speculation about the impact of hip morphological differences on behaviors like carrying seems premature to us at this stage of the work because we have not simulated any type or placement of burden. Long distance walking implies changes in kinematics or kinetics associated with fatigue or terrain characteristics (slopes, substrate evenness or compliance, etc) and we have not simulated any of those situations, either, so we do not have any data.

Reviewer 2:

This paper builds incrementally on previous works by the authors on reconstructing Australopithecine anatomy in simulation environments. Here, they focused on the effects of morphing a model of human pelvic anatomy into an australopithecine form on joint reaction forces. I like the core idea of 'what happens if we scale up an australopithecine pelvis to the size of a modern human-how does it function?'

While the main findings of the paper are adequately described, there are issues with engagement with the relevant literature and reproducibility of the results that need addressing before this manuscript is suitable for publication in Biology Open.

First we are gratified that the reviewer realized that our approach is deliberately incremental, that is, we seek to modify one aspect to understand its importance in the bigger structure.

Engagement with the literature.

Firstly, the engagement with the literature on australopithecine pelvic anatomy is very brief and requires more in-depth engagement. The use of Lovejoy's reconstruction of AL288-1 needs more justification, as the 'platypelloid' shape proposed by Lovejoy is highly controversial within palaeoanthropology. It is likely that this extreme reconstruction will have significant influence on the joint reaction forces reported here. There is no reference to alternative reconstructions such as Haeusler and Schmidt's (1995) or Brassey et al.'s (2018), both of which resulted in larger dimensions.

The auricular surface and retroauricular region (iliac tuberosity) of the AL 288-1a0 left ilium are broken and dislocated as discussed in the original description “...the auricular region has been most dramatically bent posteriorward...” (Johanson et al 1982) and in the reconstructions by Häusler and Schmid (1995) and Lovejoy (1979, in an abstract for that year’s annual meeting of the then American Association of Physical Anthropology). The pelvic “reconstruction” by Brassey et al (2018) did not address this post-mortem deformation, nor did the small changes made by Wiseman in the 2023 paper, resulting in a reconstruction that is, at best, incomplete. Further, the orientation of the pelvis (pelvic tilt) is biomechanically improbable. Important to note, however, is that the purpose of the Brassey et al 2018 reconstruction was to allow for volume estimation to develop body mass predictions, i.e., not to use for biomechanics.

The Häusler and Schmid (1995) and Lovejoy reconstructions are different by 2 mm in the anteroposterior and 3 mm in the mediolateral dimension (p 369 of Häusler and Schmid 1995). The Häusler and Schmid reconstruction is more circular (rounder) because it is narrower mediolaterally and wider anteroposteriorly than that of Lovejoy. The reconstruction of Sts 14, another australopithecine, results in a rounder shape than either reconstruction of AL 288-1. The important issue here, though, is that either AL 288-1 reconstruction (or the reconstruction of Sts 14) is substantially different in shape from the modern human form. Our goal was to see if the shape of Lovejoy’s AL 288-1 pelvis (and femur), as a representation of australopithecines, would make human walking kinematics untenable, so choosing the reconstruction that is more different from modern humans was reasonable. Also, there is no way to know which of the Häusler and Schmid (1995) and Lovejoy reconstructions is more “correct” (or closer to the shape of the living creature). As always, more fossils would be helpful.

We added the following to the text: “Other reconstructions of A.L. 288-1 are available [22,57] as are those of other australopithecines (e.g. [22,58]). We chose to focus on A.L. 288-1 because it includes an associated femur. The Brassey reconstruction of A.L. 288-1 [57], which was developed to determine segment masses, and the adjustment made by Wiseman [6] do not reconstruct the damage noted in the original fossil description [25] to the auricular and retroauricular regions. Both the Lovejoy reconstruction and the Häusler and Schmid reconstruction do correct this damage. The Lovejoy version of A.L. 288-1 [13] is 2 mm narrower in the anteroposterior and 3 mm broader in the mediolateral dimensions than the Häusler and Schmid reconstruction [22], making the Lovejoy reconstruction more oval than that of Häusler and Schmid. That is, the Häusler and Schmid version is closer to a birth canal shape that of a modern human [22]. We suspect that in the context of gait mechanics the modest metric differences between the Lovejoy reconstruction and that of Häusler and Schmid would have little impact on muscle activations during walking relative to the modern human condition. We selected the Lovejoy reconstruction because of chronological precedence and widespread availability.”

The sacral angle used may have some issues. Firstly, there is disagreement in the literature as to how this is measured, and the authors need to be clear as to how their angle is derived. The recent reconstruction by Wiseman (2023) is critiqued for an unrealistic sacral angle (which this reviewer agrees with) but neglects to mention that this is a modified version of the Brassey et al.’s 2018 reconstruction, which is freely available. Although Brassey et al. (2018), did not report the sacral angle, it would be simple for the authors to measure and report this (even if they disagree with the reconstruction). The article cited in text by the authors does not appear to actually have reported the sacral angle, so this citation may require revision. An in-depth review of os coxa morphology in australopithecines which may assist with further clarification is that by Crompton et al. (2024).

Thank you for catching the incorrect citation; it should have been to Kozma et al. Sacral angle is used simply as a way to quantify pelvic orientation and, of course, is only useful if the auricular surfaces of the innominates and sacrum are properly aligned. Our wording was not as precise as it should have been. We have estimated the orientation of Brassey et al 2018 reconstruction and included those estimates.

The literature review of multibody dynamic simulation in studies of extinct animals needs more reference to more recent work and reviews (e.g. Demuth et al., 2023, Brassey et al., 2018, Bates et al., 2025).

Added in several places.

Page 4, line 3-23 needs revision to take account recent work by Wiseman (2023) and Bates et al. (2025).

Added.

Reproducibility of analyses.

All transformations of models are described in very general terms and the readers are referred to scripts which are not provided for exact methodologies. The underlying 3d models analysed in this MS are not provided-this is necessary for people to be able to properly evaluate the findings, especially when submitting to a Journal titled 'Biology Open' and would be in line with best practice in the field.

We are happy to provide the landmark coordinates that are utilized in the TPS warping of the AnyBody modern human-like pelvis to the australopithecine-like one. The files are now provided in the Supplementary Information. Note that these files contain scaled (to the appropriate segment dimensions to match the particular subject) data. Also, the Sylvester and Kramer 2023 publication provides much more detail on the extensive steps involved in the transformation.

Other points

Page 12 lines 12-19. This is unclearly written and needs revising.

Changed to: “To preserve the integrity of the parameterization (described above), we maintained the hip joint locations which preserves the hip breadth that was determined for each human subject. That is, while the pelvis and proximal femur were morphed to reflect the shape of australopithecine-like morphology, these elements were scaled to have distances between joint centers (i.e., hip to hip distance, hip to knee distance) that were identical to the ADL modern human-like model. Finally, relative to the ADL modern human-like model, we did not change the lumbar spine and structures superior to it nor the location of the knee and morphology distal to it.”

Page 12 lines 37-44. I was under the impression that certain aspects of Australopithecine pelvis have muscle attachment sites which differ considerably in volume to modern humans (even if scaled up) and that the position of these differed somewhat. This means that the model may superficially look like an australopith's but doesn't really bear a biological resemblance. *We are aware of no hip muscle origin or insertion species-average locations that differ substantially among hominids (= humans and chimpanzees (Morimoto et al 2011; Swindler and Wood 1973)). We also agree that much caution must be used in inferring muscle attachment from bone “scars” in the absence of dissection (e.g. Wiseman 2023; Morimoto et al 2015; Suwa et al 2012), so evaluating the fossil evidence cannot provide a full picture of muscle attachments.*

Given the obvious impossibility of obtaining muscle volumes from cadavers in fossil species, the sensitivity of muscle and joint force to variation in muscle volumes could be another study, but we are skeptical that the result would be useful because there is no way to assess the validity of the muscle volumes. Wiseman 2023 attempted to estimate muscle volume by creating muscles (as volumes with polygonal shapes) that connected muscle origins and insertions (based on modern humans), and she scaled the cross sectional area of the muscle belly based on “diaphyseal shaft diameter.” Her approach is not unreasonable, but whether or not it produces muscle volumes that are more “correct” than any other method is unknowable.

To be clear, the relative location of muscle attachments may change because the shape of the bone changes. For example, if the ilium is more flared over the proximal femur, the origin of gluteus medius is more lateral relative to that of gluteus minimus than it would be in a less flared ilium. This relative location difference is due to the shape of the bone changing and not to the attachment site moving on the bone.

We also want to acknowledge, though, that we do not pose a biological question in the sense of an examination of a living creature. We do not wish to imply that our model is representative of something that lived/moved, but rather that if it had, what would the consequences be for it?

References

- Bates, K. T., McCormack, S., Donald, E., Coatham, S., Brassey, C. A., Charles, J., ... & Sellers, W. I. (2025). Running performance in *Australopithecus afarensis*. *Current Biology*, 35(1), 224-230.
- Brassey, C. A., Maidment, S. C. and Barrett, P. M. (2017). Muscle moment arm analyses applied to vertebrate paleontology: a case study using *Stegosaurus stenops* Marsh, 1887. *J. Vertebr. Paleontol.* 37, e1361432. <https://doi.org/10.1080/02724634.2017.1361432>
- Brassey, C. A., O'Mahoney, T. G., Chamberlain, A. T., & Sellers, W. I. (2018). A volumetric technique for fossil body mass estimation applied to *Australopithecus afarensis*. *Journal of Human Evolution*, 115, 47-64.
- Crompton, R., Elton, S., Heaton, J., Pickering, T., Carlson, K., Jashashvili, T., ... & McClymont, J. (2024). Bipedalism or bipedalisms: The os coxae of StW 573. *Journal of Anatomy*.
- Demuth, O. E., Herbst, E., Polet, D. T., Wiseman, A. L., & Hutchinson, J. R. (2023). Modern three-dimensional digital methods for studying locomotor biomechanics in tetrapods. *Journal of Experimental Biology*, 226(Suppl_1), jeb245132.
- Häusler, M., & Schmid, P. (1995). Comparison of the pelves of Sts 14 and AL288-1: implications for birth and sexual dimorphism in australopithecines. *Journal of Human Evolution*, 29(4), 363-383.
- Wiseman, A. L. (2023). Three-dimensional volumetric muscle reconstruction of the *Australopithecus afarensis* pelvis and limb, with estimations of limb leverage. *Royal Society Open Science*, 10(6), 230356.

Second decision letter

MS ID#: bio.061931

MS TITLE: Hip stabilization in an australopithecine-like hip: the influence of shape on muscle activation

AUTHORS: Adam David Sylvester and Patricia Ann Kramer

I am happy to tell you that your manuscript has been accepted for publication in *Biology Open*, pending our standard publication integrity checks.